# The Financial Impact of an Employee Wellness Program Focused on Cardiovascular Disease Risk Reduction

**DOI:** 10.3390/healthcare12232358

**Published:** 2024-11-25

**Authors:** Irena Boyce, Jason DeVoe, Lisa Norsen, Joyce A. Smith, Elizabeth Anson, Holly A. McGregor, Renu Singh

**Affiliations:** 1UR Medicine Quality Institute, University of Rochester Medical Center, Rochester, NY 14642, USA; 2UR Medicine Employee Wellness, School of Nursing, University of Rochester Medical Center, Rochester, NY 14642, USA; jason_devoe@urmc.rochester.edu (J.D.); lisa_norsen@urmc.rochester.edu (L.N.); joycea_smith@urmc.rochester.edu (J.A.S.); elizabeth_anson@urmc.rochester.edu (E.A.); holly_mcgregor@urmc.rochester.edu (H.A.M.); renu_singh@urmc.rochester.edu (R.S.)

**Keywords:** employee health, workplace wellness programs, cost-savings, return on investment, cardiovascular disease

## Abstract

Background. Evidence for the effectiveness and cost-savings of workplace wellness programs (WWPs) is varied, likely due to the variability in program design, as not all WWPs meet the five-point criteria of a “comprehensive WWP” set by the U.S. Centers for Disease Control and Prevention. A 2019 study of changes in cardiovascular disease (CVD) risk for those enrolled in a comprehensive WWP found that nearly half of enrolled employees with moderate to high CVD risk improved their risk compared to the initial predictions. This study extends those findings by evaluating the cost-savings and return on investment (ROI) resulting from participants’ CVD risk reduction from the employer’s perspective. Method. Cost-savings related to CVD risk were extrapolated using two studies that provided associated cost-savings for individuals participating in a WWP. Utilizing reference groups used in previous studies, we calculated cost-savings per 1% reduction in CVD risk using our population’s specific CVD risk and our program-specific costs. The cost-savings were annualized per person within each risk category. Results. Across all risk categories, cost-savings were USD 1224 per individual or USD 4.90 ROI for every USD 1 spent. Those at risk had a higher ROI per USD 1.00 spent (USD 35.4 and USD 19.2 for males and females, respectively) than those with minimal risk. However, even those with minimal risk showed a positive ROI (USD 0.3 and USD 5.0 for males and females, respectively). Conclusions/Application to Practice. Investment in WWPs should prioritize programs that include all five elements of the standards established by the U.S Centers for Disease Control and Prevention. Well-designed and comprehensive WWPs can effectively impact employee health and lead to a positive ROI and cost-savings for employers.

## 1. Introduction and Background

Workplace Wellness Programs (WWP(s)) have gained in popularity among employers with the introduction of the Affordable Care Act and the rising cost of health care [1]. Workplace wellness programs are a 50-billion-dollar industry, requiring a significant financial, talent, and time investment by employers. The success of these WWPs in improving the health of employees and reducing health care costs has been much debated over the years. Some studies demonstrate the effectiveness of WWP(s) [2,3] while others have refuted their benefit [4,5]. Such differences are likely related to the variability in program design, as not all WWPs meet the criteria [6] of a “comprehensive WWP” [2,7].

According to Healthy People 2010, in order to be considered “comprehensive”, a WWP should consist of five elements: (1) individualized heath education that includes behavior changes and lifestyle management information; (2) an organizational culture and offerings that promote health and health behavior changes; (3) full integration of the WWP into the organizational structure; (4) cross-referrals between the WWP and employee assistance programs (EAP(s)); and (5) biometric screening programs with referrals to medical care follow-up, when necessary [6]. Additionally, WWPs that offer an individualized approach to wellness that is specific to a person’s disease state, and do not try to implement “a one size fits all approach”, may be the most cost-effective approach [7,8]. The Workplace Health in America 2017 survey found that while 46% of worksites offered a WWP, only 11.8% of worksites contained all five elements of a comprehensive WWP [6].

The Colorado Heart Healthy Solutions program found a significant reduction in 10-year risk for coronary heart disease using the Framingham Risk Score [9], and Smith and colleagues [10] examined the associated cost-effectiveness among the overall program population and the at-risk participants, and found a greater ROI among at-risk participants. The University of Rochester established a personalized and nursing-led WWP, known as UR Medicine Employee Wellness Program (UR Wellness). Currently, UR Wellness provides services to over 58,000 employees in 72 organizations across Western New York. Based on the Healthy People 2010 definitions provided above, UR Wellness meets the criteria of a comprehensive WWP. A previous evaluation of the UR Wellness program found a reduction in 10-year cardiovascular disease (CVD) risk. This evaluation found that, of the employees with a moderate to high CVD risk at baseline, 48% improved their risk compared to the predicted risk and 33% improved by a full category [11]. The purpose of this study was to evaluate the cost-savings and potential return on investment (ROI) associated with the CVD risk reduction found in our prior work.

## 2. Methods

UR Wellness Workplace Wellness Program. The UR Wellness WWP included the following components: (1) a personalized health assessment (PHA) with both a lifestyle and behavioral survey; (2) point-of-care biometric screening, which includes individualized health coaching, provided by a registered nurse at the time of the screening; (3) the Wellness Engagement Plan, a web-based educational portal that is personalized for each participant and includes a 10-year CVD risk score; (4) group and individual Wellness Coaching Programs for lifestyle and chronic disease management, including cross-referrals to medical care providers; and (5) referrals to behavioral/mental health services and Employee Assistance Programs (for a full description of the UR Wellness Program, see Pesis-Katz et al.) [11].

Program Participants. A 5-year retrospective study was conducted in 2020 that included those who participated in the wellness program between 2013 and 2017 for more than one year to evaluate the impact of the UR Wellness program on participants’ CVD risk change [11]. Key indicators of CVD risk were collected and analyzed based on the Framingham Risk Methodology for 9116 employees who voluntarily participated in the UR wellness program. Participants included individuals from both the health care and education workplace sectors. This study met the federal and University criteria for exemption by the UR Office of Subject Protection.

Measures. The current study uses the data of our prior five-year retrospective analysis to evaluate the cost-savings resulting from offering the UR Wellness WWP to employees [11]. Specifically, we extracted self-reported disease state, CVD history, and behavioral risk factors, as well as biometric data (total cholesterol, HDL, LDL, Triglycerides, blood pressure, blood glucose, height, weight, and abdominal girth). A Framingham CVD risk score was calculated and used to estimate each participant’s 10-year risk of developing CVD. The Framingham score is based on non-modifiable risk factors (sex, age, and medications for hypertension) and on modifiable factors (smoking status, total cholesterol, HDL, and blood pressure). When coupled with additional factors, such as CVD history, the score can be mapped to four risk categories (minimal, moderate, high, and very high risk). Our previous analysis [11] used these four risk categories, while Krantz et al. [9], and subsequently, Smith et al. [10] used a continuous measure of risk, with at-risk individuals defined as having a 10-year Framingham Risk Score (FRS) of 10% or greater, and if an individual reported a history of coronary heart disease, an additional 10 percentage points were added to their score. Consequently, we reclassified our at-risk group to match that of Smith, which led to a different distribution of individuals across the risk categories in our sample when compared to our previous work [10,11]. Additionally, Smith et al. [10] used the term “base case” to refer to minimal-risk groups. Here, we refer to these individuals as “minimal risk”.

Cost-savings and ROI Calculations. To calculate the ROI, we used the perspective of the self-insured employer and a time horizon of 1 year. We calculated the ROI by dividing (overall annualized net cost-savings per person)/(annual program costs per person). Cost-savings were extrapolated to our population based on Smith [10], who used claims data to calculate the ROI. Smith and colleagues [10] applied a statistical model to the CHHS [9] program data in which probabilities for a cardiac event(s) were calculated based on CVD risk. These two studies, combined, provided the associated cost-savings related to CVD risk score(s) reduction for individuals who participated in the employee wellness program, compared with individuals on the same health plan that did not participate in the wellness program. Cost-savings were calculated related to CVD risk reduction for the following reference groups: (1) females at risk for CVD; (2) males at risk for CVD; (3) females at minimal risk for CVD; and (4) males at minimal risk for CVD. The author’s definition of at-risk included individuals with an FRS of greater than or equal to 10%; the minimal-risk groups included individuals with an FRS of less than 10%. We used the two CHHS findings referenced above [9,10] as we did not have access to claims data for our own population. Using these two references’ cost-savings and percent risk reduction, we calculated the cost-savings per 1% reduction in risk. The calculated cost-savings per 1% reduction in CVD risk were applied to the reclassified reference groups in our UR Wellness study population, using our study population’s specific CVD risk reduction (Table 1).

The annual cost-savings were calculated for each person within each risk category of the reference groups. The cost-savings analysis was performed from the perspective of a self-insured employer offering the UR Wellness Program to their employees and were based on the incurred employer costs of annual participation in any or all program offerings. Since UR Wellness is a comprehensive WWP, each participant had the opportunity to engage in all components of the program. We took a conservative approach in calculating our program’s costs to employers by including all the costs of the programs offered to the employees. For each of the four new groups of no-risk/at-risk females and no-risk/at-risk males, we calculated the actual program costs based on the offerings that those individuals chose to participate in, as these would incur additional costs to the employer. The specific cost components that were included are as follows: on-site biometrics screening with point-of-care results and coaching; educational materials provided at the time of screening; wellness website that includes a personal health assessment questionnaire, summary results with individualized lifestyle recommendations, targeted educational content, interactive educational modules, and personalized outreach into programs; communication with primary care providers; individualized one-on-one coaching with a multidisciplinary nurse-led team to help participants manage their chronic conditions and improve overall quality of life. The main driver for the difference in cost between the groups of participants was the utilization of individualized coaching.

## 3. Results

Table 1 describes the demographic characteristics of all study participants, based on their baseline risk category (at risk for CVD; at minimal risk for CVD). Overall, 9116 individuals participated in the program for more than one year and provided complete data for CVD risk. The majority of the population were married or living as married, white/Caucasian, non-Hispanic, and had a bachelor’s degree or higher. The average years of program participation was slightly over 3 years. The number of individuals at minimal risk for CVD was 8804 (96.6% of all participants), and the majority were females (69%). The number of individuals at risk for CVD was 312, with the majority being males (81.4%). On average, the group of individuals at risk for CVD were older compared to those with minimal risk for CVD by 15 years (for females) and by 20 years (for males).

Table 2 presents the results from our ROI analysis and the estimated cost-savings related to CVD risk score(s) reduction. When these savings are combined based on the number of participants in each category, the overall annual cost-saving is USD 1224.23 per individual or USD 4.90 ROI for every USD 1 spent. Both at-risk groups (males and females) demonstrated the highest annualized cost-savings per person, with an ROI of USD 35.40 (males) and USD 19.20 (females) for every USD 1 spent. Males and females with minimal CVD risk, despite showing a lower cost-savings for program participation, still demonstrated a positive ROI of USD 0.30 (males) and USD 5 (females) for every USD 1 spent.

## 4. Discussion

Not all WWPs are designed the same, nor do they all provide the same value. Investment in well-designed, comprehensive WWPs that lead to a reduction in CVD risk can also produce a significant ROI. The UR Wellness program demonstrated, in an observational study, not only a significant reduction in CVD risk for participants, but also a substantial ROI from the employer perspective of USD 4.90 being saved for every USD 1 spent over a five-year period.

Well-designed research is important for understanding the effects of interventions or initiatives in all areas of medical care. While randomized controlled trials (RCT) are the “gold standard” of research, in the case of ongoing programs aimed at improving health, RCTs are not practical, and in some cases are not ethical [12]. In fact, many of the WWPs evaluated in RCTs were offered to all employees as template-based programs that were not customized to employee needs and did not meet the Healthy People 2010 or the CDC criteria for a comprehensive WWP [8]. Our results are based on an observational study design, and although this can be a limitation, observational study designs allow for the wellness programs to operate without disruption from research personnel and preserve the effects that naturally occur because of the intervention. As the former CDC director suggested, “There will always be an argument for more research and better data…the goal must be actionable data—data that are sufficient for clinical and public health action that have been derived openly and objectively and that enable us to say, ‘Here’s what we recommend and why’” [12].

Our study examined the program’s impact on those who volunteered to participate in a WWP. Some have argued that this leads to self-selection bias [8], and we acknowledge this as a potential limitation. It is important, however, not to conflate evaluating the effectiveness of a WWP with investigating the feasibility of having all employees enroll in a WWP. Although we would like to eliminate self-selection bias, WWPs necessarily attract individuals interested in, and motivated to work on, their health. Failure to document the observed effects of WWPs because of this bias effectively minimizes the impact of the WWPs on participating employees and employers at large. Thus, there is value in studying WWPs in situ to understand the impact of the WWP in the real world.

Claims data were not available for our study population, which required us to evaluate ROI using similar population estimates from other studies. The intervention in the reference studies used for the ROI calculations [9,10] was different from the intervention in our study. In addition, the average age of our study participants was younger compared to the reference study. However, we were able to isolate the impact of a 1% reduction in CVD risk and its impact on costs. We applied that cost estimation to our specific CVD risk reduction, making the comparison possible. We applied the cost and applicable savings to our study population, taking a conservative approach in estimating the overall impact of the program.

Future research should focus on using objective data, such as claims, to evaluate the impact and ROI of WWPs on participants. Also important for future research is an extension of the study time frame so that the longer-term impacts of a WWP on CVD risk can be determined.

## 5. Conclusions/Implications

The established standards, provided by Healthy People 2010 for a comprehensive WWP, can offer a framework for evaluating WWPs in the context of their effectiveness. We should not expect WWPs that do not include all five elements to be as effective in impacting employees’ health outcomes, which then lead to a positive ROI. Any investment in WWPs should prioritize those programs that are comprehensive, individualized to employee’s specific needs, and well-designed to produce any impact on employees’ health and provide cost reductions for the employer organization.

## Figures and Tables

**Table 1 healthcare-12-02358-t001:** Baseline demographic characteristics for ‘No Risk’ and ‘At Risk’.

Demographic Characteristic	‘No Risk’ Group		‘At Risk’ Group	
	Male	Female	Male	Female
Total number of participants	2723 (30.9%)	6081 (69.1%)	254 (81.4%)	58 (18.6%)
Age	39.7 ± 11.0	41.2 ± 12.0	59.5 ± 7.2	56.2 ± 6.6
Marital status				
Married or living as married	2120 (77.9%)	3867 (63.6%)	223 (87.8%)	35 (60.3%)
Widowed	4 (0.2%)	70 (1.2%)	6 (2.4%)	3 (5.2%)
Divorced	130 (4.8%)	779 (12.8%)	17 (6.7%)	14 (24.1%)
Single, never married	469 (17.2%)	1363 (22.4%)	8 (3.2%)	6 (10.3%)
Race				
White/Caucasian	2204 (80.9%)	4969 (81.7%)	230 (90.6%)	46 (79.3%)
Black/African American	107 (3.9%)	429 (7.1%)	10 (3.9%)	10 (17.2%)
Asian	256 (9.4%)	360 (5.9%)	9 (3.5%)	0 (0.0%)
Other/multi/not stated	156 (5.7%)	323 (5.3%)	5 (2.0%)	2 (3.5%)
Ethnicity				
Percent Hispanic	96 (3.5%)	243 (3.6%)	2 (0.8%)	1 (1.7%)
Educational level				
Less than high school	15 (0.6%)	21 (0.4%)	6 (2.4%)	0 (0.0%)
High school grad	175 (6.4%)	329 (5.4%)	33 (13.0%)	6 (10.3%)
Bachelor’s degree	1415 (52.0%)	3886 (63.9%)	136 (53.5%)	48 (82.8%)
Master’s degree or higher	1118 (41.1%)	1843 (30.3%)	79 (31.1%)	4 (6.9%)
Total years of program participation	3.1 ± 1.1	3.3 ± 1.2	3.4 ± 1.2	3.2 ± 1.1

**Table 2 healthcare-12-02358-t002:** Cost-savings and Return on Investment.

Risk Category	Annualized Cost-Savings per Person (a)	Program Cost per Person (b)	Return on Investment (c) = (a − b)/b
No-risk males (*n* = 2723)	$246.40	$188.99	0.3
No-risk females (*n* = 6081)	$1268.40	$211.02	5.0
At-risk males (*n* = 254)	$9579.50	$263.21	35.4
At-risk females (*n* = 58)	$5909.99	$293.02	19.2
Total	$1224.23	$206.42	4.9

## Data Availability

The data presented in this study are available on request from the corresponding author. These data and associated analyses were collected for quality improvement purposes and cannot be shared for privacy and ethical reasons.

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
