# Peer review of "The Financial Impact of an Employee Wellness Program Focused on Cardiovascular Disease Risk Reduction"

_healthcare, 2024, doi:10.3390/healthcare12232358_

Round 1

Reviewer 1 Report

Comments and Suggestions for Authors

A good example of a cost-effectiveness study. It would be useful to explain some points.

In which sectors was the program applied to employees?

What type of epidemiological study should be written?

There are gender and age group differences in the risk group. What advantages and disadvantages does the intervention create in terms of effectiveness?

What is the employee group's frequency of work accidents and occupational diseases? Did the intervention reduce the risk of occupational diseases?

What is the frequency of acute coronary syndrome among participants? How much did the intervention program reduce the frequency of diseases?

Author Response

Reviewer 1 marked “Can be improved” for all general evaluation questions.  We are confident that the changes made, and detailed below, will improve those ratings.

Comment 1: In which sectors was the program applied to employees? 

Response 1: We thank the reviewer for this comment and have added a sentence (line numbers 86-87) that describes the work sectors in which our participants were employed. 

Comment 2:  What type of epidemiological study should be written?

Response 2: An epidemiological study is beyond the scope of this paper.  However, we would like to refer the reviewer to our prior study that published the impact of our clinical intervention on participants’ outcomes.  This is mentioned on lines 63-67.

Comment 3: There are gender and age group differences in the risk group. What advantages and disadvantages does the intervention create in terms of effectiveness? 

Response 3: We thank the reviewer for this question.  We would like to clarify that both sex and age are risk factors in the calculation of the Framingham CVD Risk Score, as further explained on lines 93-97 in the paper.  The effectiveness analysis based on different individual characteristics is described in our prior work and we are applying these findings in this paper to further estimate the cost implications of a WWP.

Comment 4: What is the employee group's frequency of work accidents and occupational diseases? Did the intervention reduce the risk of occupational diseases?

Response 4: While understanding the frequency of work accidents and occupational diseases is an important question, this is beyond the scope of the current paper.

Comment 5: What is the frequency of acute coronary syndrome among participants? How much did the intervention program reduce the frequency of diseases?

Response 5:  Despite the fact that we had a long follow up (5 years), this is still not long enough to develop coronary syndrome symptoms.  Thus, we used the validated Framingham Risk Calculator to determine the risk of developing CVD.

Reviewer 2 Report

Comments and Suggestions for Authors

The manuscript is aimed to answer a very relevant questions of economic effectiveness of WWPs, however, there are some issues that if resolved, would significantly increase the usefulness of the paper for readers.

- Clarification of time frame for program participants, lines 77-81: a 5-year retrospective study was conducted in 2020 that included those who participated between 2013 to 2017 for more than 1 year.

- Clarification on measures, lines 92-99: it appears that the authors used their previous study and recalculated major outcome variable using a different methodology. Please clarify the additional 10 points due to history of CHD; Smith et al. seems to limit definition of risk categories as "We defined at-risk participants as any individual with an uncontrolled risk factor or having a Framingham Risk Score of 10% or greater"(p.3) and "Recurrent CVD events are not used because these events in those with incident (new) CVD are relatively low in contemporary practice"(p.4). Since scores are not presented in the paper, it is not clear how the change in score affected the grouping and outcomes; the sample size decreased from 16000 to 9116 and the excluded population may differ from those who were used in the study calculations.

Cost-saving and ROI Calculations (line 101); there are several questions:

- Please clarify how the costs were calculated from the perspective of employer. Lines 124-133 describe the program components. A short explanation how the 38-39% increase is cost for at-risk participants was defined would be useful.

- Please clarify how the savings were calculated from the perspective of the employer. Smith et.al., used claims data (although did not specify the source of data), which may or may not be comparable to UR Wellness costs. Depending on health plan, these cost may or may not affect an employer's $1.8 million program expense. For example, third party insurance would limit cost saving for the employer, while HMO-like system may enable the researchers to access direct savings instead of using probability approach.

- Smith et.al, calculations used a scenario of 52-year-old participant, while average age of the participants in the current study is approximately 46 years. Please justify the appropriateness of savings extrapolation.

- Please clarify how cost savings per 1% reduction in CVD risk was calculated (line 117).

While the study answers an important question of cost effectiveness of WWPs, providing additional details would improve the usefulness of this research to wide audience of interested readers.

Author Response

We thank the reviewer for their thoughtful consideration of this paper. Reviewer 2 noted in the general evaluation questions that the design of the study and the description of the methods “Must be improved” and that the Introduction and the Results “Can be improved”.  We feel the changes made to the manuscript, with the suggestions of the reviewer, strengthens the paper and addresses the reviewer concerns.

Please find detailed responses to each of Reviewer 2’s comments below:

Comment 1: Clarification of time frame for program participants, lines 77-81: a 5-year retrospective study was conducted in 2020 that included those who participated between 2013 to 2017 for more than 1 year.

Response 1: We thank the reviewer for the suggestion to clarify the time frame.  We have made this change throughout the paragraph, lines 80-83.

Comment 2: Clarification on measures, lines 92-99: it appears that the authors used their previous study and recalculated major outcome variable using a different methodology. Please clarify the additional 10 points due to history of CHD; Smith et al. seems to limit definition of risk categories as "We defined at-risk participants as any individual with an uncontrolled risk factor or having a Framingham Risk Score of 10% or greater"(p.3) and "Recurrent CVD events are not used because these events in those with incident (new) CVD are relatively low in contemporary practice"(p.4). Since scores are not presented in the paper, it is not clear how the change in score affected the grouping and outcomes; the sample size decreased from 16000 to 9116 and the excluded population may differ from those who were used in the study calculations.

Response 2: We believe that the reviewer is referring to the Krantz et. al,. article, page 3, which they quoted in their comment. Page 4 indicates “If a participant reported a personal history of CHD, 10% was automatically added to the calculated FRS” (p.4).  We clarified the description of how our CVD risks scores were calculated to match those of Krantz and Smith. We do note, however, that there was no change or decrease in sample size due to this reclassification. The final sample size for both our original effectiveness study as well as the present paper includes the same sample size of 9116 participants.  These changes can be seen on lines 98-105.

Comment 3: Cost-saving and ROI Calculations (line 101); there are several questions:

Comment 3A: Please clarify how the costs were calculated from the perspective of employer. Lines 124-133 describe the program components. A short explanation how the 38-39% increase is cost for at-risk participants was defined would be useful.

Response 3A: We thank the reviewer for this comment and have added a clarification on lines 138-141 and lines 148-149.

Comment 3B: Please clarify how the savings were calculated from the perspective of the employer. Smith et.al., used claims data (although did not specify the source of data), which may or may not be comparable to UR Wellness costs. Depending on health plan, these cost may or may not affect an employer's $1.8 million program expense. For example, third party insurance would limit cost saving for the employer, while HMO-like system may enable the researchers to access direct savings instead of using probability approach.

Response 3B: We clarified on lines 110 and 132 that we are discussing the costs relative to a self-insured employer, which means that all costs for the program, and all savings, were incurred by the employer.

Comment 3C: Smith et.al, calculations used a scenario of 52-year-old participant, while average age of the participants in the current study is approximately 46 years. Please justify the appropriateness of savings extrapolation.

Response 3C: Taking this approach is actually conservative because our population is younger than those in Smith et.al. and therefore, will have lower healthcare costs.   Lines 208-209 and 211-213 further clarifies this point.

Comment 3D: Please clarify how cost savings per 1% reduction in CVD risk was calculated (line 117). While the study answers an important question of cost effectiveness of WWPs, providing additional details would improve the usefulness of this research to wide audience of interested readers.

Response 3D: We thank the reviewer again for this comment.  This question is addressed in the paper on lines 109-127 and 138-141 and we have added some clarifications to address it further.

Reviewer 3 Report

Comments and Suggestions for Authors

Dear Authors,

It is synthetic article. Interesting topic.

The purpose of the study has been clearly specified as (lines 64-66) "to evaluate the cost savings and potential return on investment (ROI) associated with the CVD risk reduction found in our prior work". 

The introduction provided sufficient information to this topic.

Methodology is clearly presented and explained. This same with the results. 

Discussion applies to all results. However, there is not information on the limitations of this study as well as the direction of future research. 

References are quite limited to 12 positions however it is the result of the type of conducted research. 

So, I would consider to provide info on the limitations of research and also try to indicate the direction of future research.

Author Response

We thank Reviewer 3 for their helpful feedback.  Reviewer 3 indicated that the results and conclusions “Can be improved” in the general evaluation questions.  We believe that our detailed response to Reviewer 3’s comments below will address these concerns.

Comment 1: It is synthetic article. Interesting topic.

The purpose of the study has been clearly specified as (lines 64-66) "to evaluate the cost savings and potential return on investment (ROI) associated with the CVD risk reduction found in our prior work". 

The introduction provided sufficient information to this topic.

Methodology is clearly presented and explained. This same with the results. 

Discussion applies to all results. However, there is not information on the limitations of this study as well as the direction of future research. 

Response 1: We address limitations in the context of our discussion and would like to point out that we do this on lines 188-191 and 197-198.  We have also added the direction of future research on lines 214 to 217.

Comment 2: References are quite limited to 12 positions however it is the result of the type of conducted research. 

Response 2: Thank you for this comment.  We used the number of references necessary for the arguments in the paper.

Comment 3: So, I would consider to provide info on the limitations of research and also try to indicate the direction of future research.

Response 3: We have addressed this in the response to Reviewer 3’s Comment 1.

Reviewer 4 Report

Comments and Suggestions for Authors

Value for money studies are critical to understanding the effectiveness of interventions, in this case WWP's.

The research is carefully described and the result clearly presented. Overall an interesting report.

Please can you check the abstract and decrease the acronyms - I was not sure what CDC stands for. It would be worth proof reading the whole report for acronyms that are not explained. A sentence beginning in Line 61, 'A previous evaluation...' is very hard to follow and should be rewritten.

Author Response

Comment 1:  The research is carefully described and the result clearly presented. Overall an interesting report. Please can you check the abstract and decrease the acronyms - I was not sure what CDC stands for. It would be worth proof reading the whole report for acronyms that are not explained. A sentence beginning in Line 61, 'A previous evaluation...' is very hard to follow and should be rewritten.

Response 1:  We thank the reviewer for this comment.  We have reduced the number of acronyms in the abstract, and have been sure that all acronyms have been defined before using them.  Also we have rewritten the sentence beginning on line 63 to make it easier to follow.